# Challenges in Gluten Analysis: A Comparison of Four Commercial Sandwich ELISA Kits

**DOI:** 10.3390/foods11050706

**Published:** 2022-02-27

**Authors:** Plaimein Amnuaycheewa, Lynn Niemann, Richard E. Goodman, Joseph L. Baumert, Steve L. Taylor

**Affiliations:** 1Department of Agro-Industrial, Food, and Environmental Technology, King Mongkut’s University of Technology North Bangkok (KMUTNB), Bangkok 10800, Thailand; plaimein.a@sci.kmutnb.ac.th; 2Food Allergy Research and Resource Program (FARRP), Department of Food Science and Technology, University of Nebraska, Lincoln, NE 68588-6207, USA; lniemann1@unl.edu (L.N.); rgoodman2@unl.edu (R.E.G.); jbaumert2@unl.edu (J.L.B.)

**Keywords:** gluten detection, ELISA, gluten-free, labeling, grain, food matrix

## Abstract

Gluten is composed of prolamin and glutelin proteins from several related grains. Because these proteins are not present in identical ratios in the various grains and because they have some differences in sequence, the ability to accurately quantify the overall amount of gluten in various food matrices to support gluten-free labeling is difficult. Four sandwich ELISAs (the R-Biopharm AG R5 RIDASCREEN^®^, the Neogen Veratox^®^ R5, the Romer Labs AgraQuant^®^ G12, and the Morinaga Wheat kits) were evaluated for their performance to quantify gluten concentrations in various foods and ingredients. The Morinaga and AgraQuant^®^ G12 tests yielded results comparable to the two R5 kits for most, but not for certain, foods. The results obtained with the Morinaga kit were lower when compared to the other kits for analyzing powders of buckwheat and several grass-based products. All four kits were capable of detecting multiple gluten-containing grain sources including wheat, rye, barley, semolina, triticale, spelt, emmer, einkorn, Kamut™, and club wheat. Users of the ELISA kits should verify the performance in their hands, with matrices that are typical for their specific uses. The variation in results for some food matrices between test methods could result in trade disputes or regulatory disagreements.

## 1. Introduction

Two major types of adverse immunological reactions occur from ingestion of gluten-containing grains (wheat, barley, rye, and related grains) among susceptible consumers: cell-mediated reactions and immunoglobulin E (IgE)-mediated allergic reactions [1]. Certain proteins in these grains are responsible for both types of reactions. These grains contain multiple proteins that are classified into several categories: albumins, globulins, prolamins, and larger prolamin-like glutelins. The latter two types form viscoelastic masses when mixed and kneaded with water.

The prolamins and glutelins are associated with provocation of celiac disease (CeD), dermatitis herptiformis (DH), and non-celiac gluten sensitivities (NCGSs) [1]. CeD is estimated to affect nearly 1% of the U.S. population [2]. NCGSs affect an unknown but possibly larger percentage [3]. CeD is a T-cell mediated response to digestion-resistant gluten peptides [4]. CeD is a delayed hypersensitivity reaction with symptoms typically occurring 48–72 h after acute exposures to gluten. Native gluten peptides and those deamidated by the consumer’s endogenous tissue transglutaminase mediate the enteropathy [5]. CeD and DH are provoked by all of the gluten-containing grains (wheat, rye, and barley) except oats that are tolerated by the vast majority of individuals with CeD if pure and uncontaminated with the other gluten-containing grains [6]. Individuals with NCGS experience intestinal symptoms on ingestion of gluten but do not display the intestinal inflammatory damage that is characteristic of untreated CeD [3]. The mechanism of NCGS has not yet been elucidated. CeD, DH, and NCGS are managed with gluten avoidance diets in part through the consumption of gluten-free (GF) foods.

All of the gluten-containing grains can cause IgE-mediated, immediate hypersensitivity reactions [7]. Unlike CeD, cross reactions do not occur among the various gluten-containing grains for most affected individuals. Wheat allergy is, by far, the most prevalent of the allergies to the various gluten-containing grains affecting <1% of the U.S. population [8,9,10]. Higher prevalence was reported in Thailand, South Korea, and Japan, where rice is the staple cereal [11,12,13]. Some individuals with wheat allergy experience reactions only when they exercise concomitantly with ingestion of wheat, a condition known as wheat-dependent, exercise-induced anaphylaxis (WDEIA) [7]. Individuals with IgE-mediated allergies to one of the gluten-containing grains must only avoid that single grain and any protein-containing ingredients derived from it. However, for practical reasons, these consumers also likely rely, in part, on gluten-free products in their avoidance strategy.

To protect consumers with food allergies and intolerances, the Codex Alimentarius Commission established guidance that required the declaration of certain foods or ingredients derived therefrom on food package labels in 1999 [14]. That priority list included cereal sources of gluten, defined a protein from wheat, rye, and barley. Most countries subsequently enacted regulations requiring the labeling of cereal sources of gluten. However, in the USA, wheat was recognized for its role in IgE-mediated food allergies and a regulation was established to require source labeling of wheat and any ingredients derived therefrom on packaged food labels [15]. Barley, rye, and other gluten-containing grains and their ingredients were not included in this regulation.

Subsequently, the Codex Alimentarius Commission proposed a global Standard for Foods for Special Dietary Use for a Person Intolerant to Gluten in 2008 [16]. According to this standard, a food product would be considered gluten-free only if the final gluten level in the product is determined to be less than 20 ppm gluten, defined as those proteins, commonly found in wheat, triticale, rye, barley, or oats to which some consumers are intolerant. Many countries, including the U.S., have developed regulations for gluten-free labeling that comply with the recommended level of 20 ppm gluten. However, some countries have also allowed the marketing of oats as gluten-free if residual gluten levels are <20 ppm [17].

The global implementation of gluten-free labeling regulations requires the use of methods for the detection of gluten residues from wheat, rye, barley, and related grains at a level of sensitivity below 20 ppm gluten. To assure compliance with gluten-free labeling regulations and to protect sensitive consumers, the food industry must ensure the absence of gluten (down to <20 ppm) by careful formulation, control of ingredient sources, segregation of the manufacture of gluten-containing foods, and application of accurate labeling. The success of gluten-free labeling relies on the accuracy and limitations of the analytical methods used to detect gluten residues. Multiple, quantitative commercial methods, primarily enzyme-linked immunosorbent assays (ELISAs), are available for the detection of gluten residues in foods [18,19,20]. ELISA-based methods are relatively simple, fast, and have become the most widely used gluten detection methods. ELISA-based methods have been reported to be reliable in detecting gluten at the level of 20 ppm in a variety of types and forms of food [21]. However, only one of these ELISA methods, the R5 Méndez method has been recognized as a Type I method (defining method) for gluten analysis, an AOAC Official Method^SM^ [22]. In general, it is permissible to use equivalent methods for the detection of gluten residues. The U.S. FDA regulation does not specify any specific analytical methods for compliance testing but recommends that food manufacturers may choose any scientifically valid method that is most appropriate to reliably detect the presence of 20 ppm gluten in their products [23]. However, limited data exist on the equivalency of results obtained by the various commercial gluten ELISAs across a range of relevant food matrices and at low residue levels around the 20-ppm limit. The existing commercial gluten ELISA kits have variations in format, antibody, specificity, sensitivity, and extraction buffers and different results were observed in many food samples [24,25]. The consequences of any variability in results by use of different methods of analysis are obvious in terms of buyer-seller expectations and regulatory compliance.

As the choice of analytical method is crucial to ensuring consumer safety, the objective of this study was to compare the suitability and performance of four commercial sandwich ELISA kits to confirm the gluten-free status of foods and ingredients. Gluten analysis was performed on 32 foods and ingredients representing various matrices and focusing primarily on samples that had low levels of gluten residues surrounding the 20-ppm target level for gluten-free status. Additionally, the ability of these same four commercial sandwich ELISAs to detect gluten residues from a wide range of gluten-containing and other grains was evaluated.

## 2. Materials and Methods

### 2.1. Commercial Sandwich ELISA Kits and Reagents

The kits used were the RIDASCREEN^®^ Gliadin kit (R-Biopharm AG, Darmstadt, Germany; Art. No. R7001; the AOAC-RI license #120601, the AOAC-OMA license #2012.01, and the AACC International Approved Method 38–50.01), the Veratox^®^ for Gliadin R5 kit (Neogen Corporation, Lansing, MI, USA; Product No. 8510; the AOAC-RI license# 061201), the Wheat Protein ELISA kit (Gliadin) (Morinaga Institute of Biological Science, Inc., Yokohama, Japan; Cat. No. 181GD), and the AgraQuant^®^ Gluten G12 assay (Romer Labs UK Ltd., Cheshire, UK; Product No. COKAL0200; the AOAC-OMA license #2014.03 and the AACC International Approved Method 38–52.01). The Neogen gliadin renaturing cocktail solution was purchased from Neogen Corporation (Lansing, MI, USA). Ethanol was purchased from EMD Chemicals Inc. (Gibbstown, NJ, USA). Beta-mercaptoethanol was purchased from Sigma Aldrich (St. Louis, MO, USA). Skim milk powder was purchased from Oxoid Ltd. (Basingstoke, Hampshire, UK). The testing of samples with each kit was performed according to the manufacturer’s instructions. The absorbance values were read using a Spectra MR^TM^ Microplate Spectrophotometer (DYNEX Technologies, Inc., Chantilly, VA, USA). Samples with absorbance values over the highest standard were diluted further and re-tested until the values fell within the range of standards. The parameters used in each kit are summarized in Table 1.

### 2.2. The 32 Food and Ingredient Samples

Since our laboratory has analyzed a number of foods and ingredients for the food industry and CeD support organizations for many years, we selected a set of 32 samples (Table 2) to represent a wide range of grains and gluten concentrations and to encompass a wide range of food matrices. All the samples had previously tested at levels of between 5–1000 ppm gluten by the RIDASCREEN^®^ Gliadin kit or the RIDASCREEN^®^ FAST Gliadin kit, except for the raw meat sample which was previously tested to contain more than 1000 ppm gluten but was chosen to include a meat matrix. Samples were processed to provide a relatively uniform distribution of sample material prior to extraction. Raw meat was blended using a Cuisinart Mini-Prep Plus^®^ processor (Model DLC-2A). Cereal bars, chips, and cookies were milled using a commercial blender (Osterizer^®^, Boca Raton, FL, USA). Two subsamples were analyzed in duplicate using the R-Biopharm AG, the Neogen, and the Morinaga kits. Three subsamples were analyzed in duplicate with the Romer Labs kit.

### 2.3. Preparation and Analyses of Cereal Grain Spiked Samples

Sixteen powders of wheat, barley, rye, triticale, oat, and sorghum were used in the spike-and-recovery experiments (Table 3). The following samples were purchased from local stores in Lincoln, NE: C&H^®^ pure cane granulated white sugar; organic whole grain barley flour (Arrowhead Mills, Inc., Hereford, TX, USA); organic whole grain rye flour (Arrowhead Mills, Inc.); semolina wheat flour (General Mills Corp., LLC, Minneapolis, MN, USA); 100% stone ground whole (hard red spring) common wheat flour (Bob’s Red Mill, Milwaukie, OR, USA); and gluten-free, stone ground white rice flour (Bob’s Red Mill). Kiln roasted whole grain oat flour and kiln roasted rolled oats (both labeled gluten-free and wheat-free) were purchased from Château CREAM HILL Estates (LaSalle, Québec, Canada). Organic whole Durum wheat flour was purchased from Barry Farm Foods (Wapakoneta, OH, USA). Whole emmer wheat grains were purchased from Bluebird Grain Farms (Winthrop, WA, USA). Einkorn wheat kernels were purchased from InfraReady Products (1998) Ltd. (Saskatoon, Saskatchewan, Canada). Whole grains of club wheat were provided by Dr. David Jackson (Nebraska Agricultural Experiment Station). Whole grains of sorghum (*Sorghum bicolor* var. Macia) and two varieties of oat (*Avena sativa* var. Jerry and var. Ogle) grown in fields separated from wheat, barley, rye, and triticale to prevent cross-contamination were provided by the Department of Agronomy and Horticulture and Husker Genetics (University of Nebraska Agricultural Research Division), respectively. Whole grains of Kamut™ (Khorasan) wheat, spelt, and triticale were provided by Dr. James Glueck (Arrowhead Mills, Inc.). Crude wheat gluten was purchased from Sigma Aldrich (St. Louis, MO, USA; Cat # G5004-500G; Lot# SLBB0512V).

Prior to the spiking, the protein content of the 17 grains and flours was analyzed in triplicate subsamplings by the Dumas method (Table 3). The sugar, grains, and kernels were milled and homogenized to uniformity using a SPEX CertiPrep Freezer/Mill 6850 for three cycles (each for 2 min). For the powders of sorghum and oats, three subsamples were directly analyzed in duplicate by each of the four kits. Since the wheats, barley, rye, triticale, and crude wheat gluten are high in gluten proteins, we spiked each powder of the grains and crude wheat gluten into ground sugar since the sugar contained no detectable gluten or wheat protein by any of the four kits and was previously demonstrated to provide no interference with gluten analysis. The spiking is designed to represent possible contamination scenarios. For the crude wheat gluten, 100 mg powder was spiked into 99.9 g ground sugar to provide a sample of about 1000 ppm gluten. We also spiked the crude wheat gluten into each of the powders of sorghum and oats. Since gluten proteins account for nearly 80% of the protein in common wheat and barley [29], 1.116 g of each of the grain flour were spiked into 98.88 g of ground sugar to provide samples of around 1000 ppm gluten. To normalize the weight ratio, the other grain flours were spiked similarly. Spiking was conducted in a KitchenAid blade coffee grinder (model BCG100WH) for 2 min. Three independent spiking replicates were prepared for each of the grains. To assess the spiking homogeneity, four subsamples from each of the three independent spiking replications of each grain were analyzed in duplicate by the R-Biopharm AG kit. One subsample was analyzed in duplicate by the Neogen, the Morinaga, and the Romer Labs kits.

### 2.4. Statistical Analysis

For the 32 foods and ingredients, the grain samples, and the grain spiked sugar/oats/sorghum samples, the results were analyzed by one-way ANOVA or *t*-test. Multiple comparisons were conducted using Fisher’s LSD test and *t*-test mean comparisons were performed using a significance level of 0.05 using the SAS^®^ 9.3 software package. The results below the limit of quantification (BLQ) were excluded from the statistical comparisons.

## 3. Results and Discussion

For those with adverse reactions to gluten or any of the specific gluten-containing grains, strict life-long avoidance can be burdensome and difficult since the grains and their derivatives are widely incorporated in various ingredients and foods. In many food products, the presence of proteins from these sources is not obvious and consumers must rely on accurate labeling to avoid exposure. Gluten-free labeling implementation depends on the accuracy and limitations of the analytical methods used to quantify the gluten content. Commercial ELISA kits are utilized by many food companies to comply with the gluten-free labeling regulations.

### 3.1. The Gluten and Wheat Proteins Contents of the 32 Food and Ingredient Samples

First, the performances of four commercial ELISA kits were evaluated using 32 foods and ingredients (Table 2). While the source and form of the detectable gluten in these samples are unknown, they represent real food industry samples including many that had detectable gluten on initial analysis in low range (5–50 ppm) that might create issues if different analytical methods yielded variable results. The two R5 kits yielded similar results for most products as expected. The G12 kit and the Morinaga kit, though reporting result as wheat protein, not gluten, also yielded similar results to the two R5 kits for most samples but yielded substantially different results for a few samples including samples #22 (yeast extract), #24 (hemp protein powder), #25 (cookie), and #27 (bar flavor). Those differences could be caused by any one of the several reasons: (a) differences in the grain source of glutens and related proteins, (b) differences in the efficiency of extraction and detection, (c) subsampling differences with particulates, or (d) some combination. Moreover, though it is widely agreed that the effect of the food matrix on the detection can be reduced by dilution, it is impossible for the end user, to elucidate and to exclude the food matrix effect. Gluten proteins are diverse in character and content, and differences exist in the abundance of certain antibody-binding epitopes in varieties of wheat, barley, and rye compared to the mass of total grain or of total protein from these grains [30]. Since the source of the grain is not known for these 32 samples, it is impossible to ascribe the differences in results to the identity of the grain or to differentiate possible subsampling differences. Although we attempted to homogenize the samples thoroughly, there were apparent subsampling differences based on the high variation (large standard deviation (SD)) in results for replicate samples using the same assay kit in a few cases (e.g., #1 ground raw meat, #2 organic oat flakes, #22 yeast extract powder, #23 tortilla chips, #24 hemp protein powder, and #26 cocoa powder). Most of the variability in results were seen in samples where particulate contamination might be expected. There are cases where divergent, though not statistically different, results were obtained with the four kits. For sample #32 organic creamy buckwheat powder, the results with the four kits were: Morinaga kit, 1.14 ppm and 3.03 ppm; R-Biopharm AG kit, 44.9 ppm and 18.4 ppm; Neogen kit, 41.4 ppm and 11.6 ppm; and Romer Labs kit, 27.4, 24.9, and 10.5 ppm. The divergent results between the four kits could be attributed to the variances obtained within each kit and to the small number of samples (*n* = 2 and *n* = 3). The divergence within replicates conducted with the same kit could also be attributed to variances obtained within each kit but are more likely due to subsampling differences and to the small sample sizes that are insufficient for homogeneous subsampling, as previously observed by Fritz and Chen [31]. We do not know the true values expected for these samples because they were submitted from industry sources.

When analyzed with the Morinaga kit, the grass-based samples and especially the #28 organic Alfalfa grass juice powder and the #30 organic Kamut^TM^ grass juice powder yielded apparently lower results than the other kits. Those differences might reflect differences in the gluten and gluten-like proteins present in the grass-based products and that the epitopes targeted by the polyclonal antibodies in the Morinaga kit differ from those of the R5 and G12 monoclonal antibodies. Nevertheless, those cases demonstrated scenarios when the Morinaga kit cannot successfully confirm the results obtained by the R5 and G12 kits as previously observed by Scherf [32]. At least three food products (#28, #30, and #31) verified as gluten-free (<20 ppm) by the Morinaga kit could not be confirmed by the R5 and G12 kits. Although the exact gluten contents of these products are not confirmed, underestimation would pose a risk of clinical relapse while an overestimation could result in unnecessary product rejection or recalls. We note that the Morinaga kit uses a distinct extraction and calibration scheme. The kit is calibrated in units of wheat protein rather than gliadin, although it is stated in the instructions that “gliadin is used as a marker for wheat protein” (Table 1). The factor for converting gliadin into wheat protein for this kit is not defined. The nature of the immunogen (gliadin from one grain, mixture of gliadins, etc.,) used to sensitize animals to produce the polyclonal antibodies for the Morinaga kit is not known. Each of the R5 platforms uses a “gliadin” standard and employs the R5 Méndez monoclonal antibody that binds to similar native peptides on several prolamins from wheat, barley, rye, and triticale (Table 1). The R5 kit instructions call for multiplying the calculated gliadin results by a factor of 2 to estimate the gluten content of a sample. The Romer Labs platform utilizes a monoclonal antibody that is known to target five native peptides (Table 1) found in various prolamins and a gluten standard calibrated to the Prolamin Working Group (PWG) gliadin [33]. Because of the uncertainty about the consistency of standards between these kits and the specificity of the antibodies used in the kits, one might expect some differences in detection efficiency between samples. Similar to our observation, Sharma et al. [24] reported that the R-Biopharm AG and the Morinaga ELISA kits yielded different results in a number of food products, especially breakfast cereals and bars. 

Several samples with rather low detectable levels of gluten in the R-Biopharm AG kit gave apparent BLQ results in one or more of the other kits but we note that the levels of gluten in these cases are at or below 20 ppm by all kits except for five samples. First, sample #9 (malt drink), tested at 19.9 ppm and 18.9 ppm by the R-Biopharm AG kit, at 23.4 ppm and 27.4 ppm by the Neogen kit, at 17.6 ppm, 17.4 ppm, and 15.9 ppm by the Romer Labs kit, and at 21.7 ppm and 14.0 ppm by the Morinaga kit. Although relatively consistent, the results demonstrate the difficulty of testing to a fixed limit using sampling and detection methods that are reproducible, but not identical. Moreover, the case illustrates the complexity in detecting residual, immunogenic glutens since malting involves enzymatic hydrolysis of starches and seed storage proteins in barley grains. The sandwich ELISA kits are not appropriate for quantifying hydrolyzed glutens. Despite having quantified below 20 ppm by the R-Biopharm AG and Romer Labs kits, this barley-derived product can be labeled gluten-free only if adequate information regarding the gluten reduction process to meet the requirement is provided. Second, sample #10 (nutty rice bar), tested at 23.4 ppm and 22.1 ppm by the Morinaga kit while the results from the three other kits tested below 20 ppm. Third, sample #14 (sorghum powder), the R-Biopharm AG kit tested at 47.7 ppm and 25.6 ppm while results with the other three kits yielded <10 ppm gluten and BLQ for several replicates. Fourth, sample #15 (organic wheat grass juice powder), tested at 26 ppm and 26.1 ppm by the Neogen kit and at 19 ppm and 18.8 ppm by the R-Biopharm AG kit whereas the results with the Morinaga and Romer Labs kits were <10 ppm. Fifth, sample #20 (oat flour), tested at 16.1 ppm and 21.1 ppm by the G12 kit while the other kits tested at <10 ppm, indicating less likelihood of gluten contamination from Triticeae sources. This is, however, as expected since the G12 kit is known to react with certain varieties of oat [34,35]. Nevertheless, these results do not reflect poorly upon the use of the R5, G12 or Morinaga methods to support the proposed limit of less than 20 ppm as gluten-free. In most cases, the four kits provided reasonably consistent results and they can be used to ensure labeling of foods as gluten-free at a level of certainty that would protect CeD consumers. However, the five exceptions described above suggest that to confidently make a gluten-free label claim, the food industry should set a target gluten level well below 20 ppm since different ELISA kits could yield variable results around that critical regulatory level. The potential effects of subsampling differences also indicate the need for conservatism in the establishment of industry target levels. However, there should be additional consideration and testing of replicate samples, food matrix diversity, and a clear decision-making process that considers the realistic variation that occurs in testing. In a comparison of seven commercial gluten/wheat protein ELISA kits, Rzychon et al. [25] demonstrated that the measured gluten contents in food samples varied 10–20 times between the kits and only four of the 24 analyzed samples were measured as gluten-free by all seven of the kits. By spiking low to high levels of gluten and wheat flour in a cornbread matrix, Sharma et al. [36] indicated wide-ranging recoveries tested with the R-Biopharm AG, Morinaga, and Romer Labs kits. The authors also observed effects of the gluten source, the spiking level, and the matrix, including baking time, on the gluten determination.

### 3.2. Effect of Grain Sources on the Gluten and Wheat Proteins Contents

The gluten-free definition in the U.S. recognizes multiple grain sources of gluten. However, the ability of the existing ELISA kits to detect and quantify differing glutens from different grain sources has not widely been assessed. Thus, we moved forward to test a number of grains with the four kits (Table 3).

Even though we attempted to homogenize samples, there were apparent high variations in the results for replicate samples using the same kit in a number of samples. The results reflected subsampling differences. Nevertheless, adequate recoveries were obtained from the four kits in most samples. As expected, the two R5 kits, utilizing the R5 antibody that was raised against rye secalin extract [18], quantified higher apparent gluten contents for the rye and triticale samples when compared to the common wheat samples on a total weight basis. The results suggested that these cereal grains contain different numbers of highly homologous gluten proteins with identical or highly similar copies of the epitopes recognized by the antibodies as also previously observed by Ribeiro et al. [37]. In rye, triticale, and barley there are a higher number of epitopes recognized by the R5 antibody than in wheat [38]. In contrast, the epitopes recognized by the G12 antibody, which was developed against the 33 amino acid α-gliadin peptide present in common wheat [34], appears to be equivalently present in the extracts of common wheat, spelt, rye, and triticale (Table 3). The epitopes recognizable by the R5 and the G12 antibodies are likely to be present in higher amounts in the extracts derived from the hexaploid wheats than in those from the tetraploid and diploid wheat varieties. However, club wheat, a hexaploid wheat, tested comparatively low for apparent gluten content (Table 3), a result that is predictable because club wheat has been known for its low gluten content [39]. The differences noted between the grain samples in this study suggest the evolution of polyploidies that contain multiple copies of genetically diverse proteins. By comparing five sandwich ELISA kits (the R5, G12, Skerritt, pAb1, and pAb2), Lexhaller et al. [30] confirmed high variation in antibody recognition among prolamins and glutelins from wheat, barley, and rye sources. In addition, we found that the R-Biopharm AG kit and the G12 kit detected higher levels of apparent gluten content in barley than in the common wheat confirming earlier observations by Pahlavan et al. [40]. The G12 kit exhibited some reactivity, all below 20 ppm, to the four oat samples. Interestingly, the two R5 kits yielded apparently distinct levels of apparent gluten content in several samples including rye, barley, common wheat, club wheat, all the tetraploid wheats, and the crude wheat gluten. The higher gluten levels obtained by the R-Biopharm AG kit in those nine samples are, in part, possibly the consequence of the pretreatment with the cocktail solution containing a protein unfolding agent and a protein disaggregating agent prior to the ethanol extraction (Table 1). Similar to our finding, in a three-subsampling analysis of high gluten-containing commercial samples by Yu et al. [41], the R-Biopharm AG kit yielded higher results when compared to the Neogen kit, nearly two times higher for two wheat flours and 3.9, 4.6, and 5.6 times higher for instant noodle, udon, and spaghetti, respectively. Apart from the three kits, the Morinaga kit yielded comparable wheat protein contents in the samples of rye, triticale, barley, spelt, common wheat, all the tetraploid wheats, and the crude wheat protein (Table 3). Overall, when compared across the wheat samples, the club wheat and the einkorn wheat were detected at lower ppm gluten and wheat protein on a total weight basis. Lower test results from these ELISAs does not necessarily mean that the samples contain lesser amounts of the target protein(s). Moreover, this does not guarantee that the two samples present significantly lower risks to consumers with CeD. Several explanations are possible including that the two specific species contain less gluten, as known in the case of club wheat, or that they express distinct glutens that may not be extracted or detected equally by the specific antibodies. Even within a single wheat species, distinct cultivars had been demonstrated to have different gluten proteins [37]. As reported by Gianfrani et al. [42], two lines of einkorn wheat were similar in their capacity to stimulate T-cell restricted interferon-γ (IFN-γ) secretion, but only one line was reported to induce crypt hyperplasia. 

The masking or interfering effect of food matrices was demonstrated when the crude wheat gluten was spiked into three distinct raw matrices: sugar, oat flours, and sorghum flour. The four kits adequately measure the expected levels of crude wheat gluten in the sugar and four oat matrices (Table 3). However different results were obtained with the four kits in the sorghum matrix. The averaged results were 8.55 ppm by the Morinaga kit, 67.5 ppm by the R-Biopharm AG kit, 247 ppm by the Romer Labs kit, and 653 ppm, the most relevant result to the spiking level, by the Neogen kit. Since the variations in these kits are low, the heterogeneity of the spiked sample and subsampling effects are less likely. One possibility could be that the Macia variety of sorghum contains small but sufficient tannin to interfere with the extraction via tannin–protein interactions [43]. Proteins in sorghum have also been found to interfere with protein extraction via protein–protein interaction [44]. The results obtained from the Neogen kit suggest the success in incorporating the extraction additive to aid in extraction of the samples containing tannins/phenol compounds.

Taken together, all these kits have the capability to detect gluten proteins from all of the gluten-containing grains, but the quantitative differences are due to several factors. The two R5 kits utilize the same antibody that targets rye secalin but is also able to detect other closely related prolamins and glutelins from other grains. The two R5 kits use different gliadin standards and extraction schemes. There is a defect in normalizing glutens equally as each of these grains represent mixtures including variable amounts of prolamin and glutelin fractions [33]. Using reversed-phase high-performance liquid chromatography (RP-HPLC), Wieser and Koehler [45] demonstrated that for common wheat, the prolamin to glutelin ratio ranged from 1.5 to 3.1. The ratios ranged from 0.2 to 4.9 in commercial wheat starches, at 1.4 to 5 for barley, at 6.3 to 8.2 for rye, and at 4.9 to 13.9 for einkorn wheat. The ELISA kits applying the factor of 2 to adjust the results from gliadin to gluten are therefore likely to underestimate or overestimate gluten content in some grains and products. The G12 kit bases its quantification on a gluten standard calibrated to the PWG gliadin, the varying ratios of prolamin to glutelin in various gluten-containing grains thus would lead to misestimation. On the other hand, although the specificities of the polyclonal antibodies and the calibration standard used in the Morinaga kit are unknown to us, our findings indicated that the wheat proteins/related proteins in wheats, barley, and rye measured by the kit are relatively close. In summary, differences in gluten protein type, form and proportion, and differences in the content of interfering substances led to different results determined by each of the ELISA kits. Furthermore, based on varied constructed parameters, different results were obtained for the cereal grains using the four ELISA kits. The source of the gluten proteins and the degree of hazard to the sensitive individuals could not be ascertained without clinical studies. Nevertheless, the results were reasonably comparable between the four kits on most samples. The use of any of the four kits can provide data to support the conformance of gluten-free products with regulatory labeling requirements. In addition, our analysis using real-world food samples demonstrated appropriate sensitivity obtained from the four kits to measure the gluten contents in typical serving sizes of foods as previously investigated by Holzhauser et al. [46]. One drawback of the ELISA kits however is that the sample sizes are very small compared to the typical serving sizes, so subsample variability affects the reliability of results especially when particulate contamination is encountered.

## 4. Conclusions

To date no analytical method exists that can precisely measure glutens from different grain sources, forms, and food matrices. Our gluten analyses in real food products, ingredients, and raw grains indicated that the four sandwich ELISA platforms are not fully comparable but are capable and suitable in detecting and quantifying gluten proteins in various food samples. The four kits yield comparable results and could be used interchangeably in many food matrices. The suitability of each kit should be carefully considered before using the kit to verify the gluten content in a food. The users should also be aware that food can be heterogeneous and large sample and sampling sizes might help overcome the uncertainty in divergent results obtained from the kit(s). Our findings demonstrate that some variability within and between methods can be expected at gluten levels around 20 ppm. Thus, food companies making the gluten-free claims should select target levels well below 20 ppm to assure compliance. Finally, we agree that the AOAC-validated R5 Méndez method is an appropriate choice to detect and measure gluten proteins at the level of 20 ppm in various types and forms of food. We judge that any method that provides comparable results to the R5 Méndez method is also suitable for gluten analysis and could be used to document their suitability for the use to comply with the gluten-free definition.

## Figures and Tables

**Table 1 foods-11-00706-t001:** Parameters of the four commercial sandwich ELISA kits used in the study.

Sandwich ELISA Kit	R-Biopharm AG: RIDASCREEN^®^ Gliadin Kit	Neogen: Veratox^®^ for Gliadin R5 Kit	Romer Labs: AgraQuant^®^ Gluten G12 Assay	Morinaga: Wheat Protein ELISA Kit (Gliadin)
Antibody	mAb R5 (reported to recognize QQPFP, QQQFP, LQPFP, QLPFP, QLPYP, QLPTF, QQSFP, QQTFP, QQPYP, PQPFP, QQPFPQ, QQPFPL, PQQPFP, SQQPFP, QLPFPQ, QRPFAQ, QQSFPQ, and QXPW/FP) [26,27,28]	mAb R5 (reported to recognize QQPFP, QQQFP, LQPFP, QLPFP, QLPYP, QLPTF, QQSFP, QQTFP, QQPYP, PQPFP, QQPFPQ, QQPFPL, PQQPFP, SQQPFP, QLPFPQ, QRPFAQ, QQSFPQ, and QXPW/FP) [26,27,28]	mAb G12 (reported to recognize QPQLPY, QPQLPF, QPQLPL, QPQQPY, and QPELPY) [19]	pAb (gliadin as a marker for wheat protein)
Calibration Standard (Analytical Target)	Gliadin (the PWG gliadin)	Gliadin	Gliadin (the PWG gliadin)	Wheat protein (gliadin as a marker for wheat protein)
Detection Limit	3 ppm gluten	5 ppm gluten	2 ppm gluten	0.3 ppm wheat protein
Limit of Quantification	5–80 ppm gluten	5–80 ppm gluten	4–200 ppm gluten	0.3–20 ppm wheat protein
Protein Extraction	Extracted * with the Cocktail (patented) solution (Cat# R7006) (containing 250 mM β-mercaptoethanol, 2M guanidine hydrochloride, and 0.1X PBS) at 50 °C for 40 min followed by 80% ethanol at 25 °C for 60 min, centrifuged, and diluted with the sample diluent. ** Equal amount of skimmed milk is added in tannin and polyphenol containing samples (e.g. cocoa, buckwheat, and spices). Meat sample was homogenized prior to extraction as recommended in the instruction*	For non-heat processed samples, extracted with 60% ethanol plus the provided extraction additive for 10 min, centrifuged, and diluted with the provided PBS. For heat-processed or unknown commodities, extracted * with the gliadin renaturing cocktail solution (Cat# 8515) (containing 19% guanidine hydrochloride) at 50 °C for 40 min followed by 80% ethanol at 25 °C for 60 min, centrifuged, and diluted with the provided PBS. ** Sample containing tannin/phenolic compounds (e.g., cocoa and buckwheat) is added with the provided extraction additive.*	Extracted * with the provided extraction solution (containing <1% β-mercaptoethanol, 19% guanidine hydrochloride, and <1% PBS) at 50 °C for 40 min followed by 80% ethanol at 25 °C for 60 min, centrifuged, and diluted with the provided dilution buffer. ** For chocolate containing sample, equal amount of powdered fish gelatin is added.*	Extracted with provided extraction buffer containing β-mercaptoethanol at 25 °C for 12 hr, adjusted pH to neutral, centrifuged, and diluted with the provided diluents.
Result Interpretation	Gluten content is calculated by the software RIDA^®^ SOFT Win (Art. No. Z9999) using cubic spline model. Gluten content is conversed from gliadin content by a factor of 2.	Calculated by the Veratox 3.0.1 software using log/logit model. Gluten content is conversed from gliadin content by a factor of 2.	Calculated by the provided Romer Labs^®^ spreadsheet using linear model.	Calculated by the GraphPad Prism^®^ 403 software (GraphPad Prism^®^ software, Inc., San Diego, CA, USA) using linear model.

**Table 2 foods-11-00706-t002:** Analysis of gluten/wheat protein in foods and ingredients by the four commercial sandwich ELISA kits.

Sample		R-Biopharm AG RIDASCREEN^®^ Gliadin Kit (n = 2)					Neogen Veratox^®^ for Gliadin R5 Kit (n = 2)					Romer Labs AgraQuant^®^ G12 Assay (n = 3)					Morinaga Wheat Protein ELISA Kit (n = 2)				
		Gluten ± SD, ppm				%CV	Gluten ± SD, ppm				%CV	Gluten ± SD, ppm				%CV	Wheat Protein ± SD, ppm				%CV
1	Ground raw meat	1690	±	310	a	18.3	1694	±	451	a	1.7	1762	±	90.7	a	5.1	1972	±	488	a	24.7
2	Organic oat flakes	158	±	41.9	a	26.5	173	±	118	a	68.5	72.1	±	7.96	a	11.0	119	±	70.1	a	58.9
3	Corn flour	185	±	24.2	a	13.1	154	±	28.4	a, b	18.4	118	±	24.5	b	20.8	204	±	34.5	a	16.9
4	Brown rice syrup	131	±	9.50	a	7.2	92.7	±	9.48	b	10.2	128	±	15.1	a	11.8	63.9	±	17.6	b	27.5
5	Navy bean flour	86.8	±	3.50	a, b	4.0	110	±	14.4	a	13.1	65.5	±	18.2	b	27.8	70.6	±	8.33	b	11.8
6	Red lentil flour	73.6	±	9.57	a	13.0	63.8	±	5.59	a, b	8.8	49.8	±	6.97	b	14.0	47.5	±	6.19	b	54.3
7	Cereal powder	69.4	±	2.72	a	3.9	46.7	±	8.49	a, b	18.2	69.6	±	15.9	a	22.8	30.1	±	7.06	b	23.5
8	Wheat starch powder	41.0	±	0.06	a, b	0.1	21.8	±	5.23	b	24.0	19.9	±	3.55	b	17.8	53.0	±	19.1	a	36.0
9	Malt drink	19.4	±	0.69	a, b	3.5	25.4	±	2.79	a	11.0	17.0	±	0.94	b	5.5	17.9	±	5.43	b	30.4
10	Nutty rice bar	16.3	±	2.12	b	13.0	16.1	±	1.10	b	6.8	11.1	±	1.48	c	13.4	22.7	±	0.95	a	4.2
11	Ground rice crisp	14.0	±	0.44	a	3.1	15.4	±	0.25	a	1.6	9.00	±	1.01	b	11.2	12.5	±	3.75	a, b	30.1
12	Green pea flour	13.7	±	1.59	a	11.6	15.8	±	0.39	a	2.5	11.4	±	2.58	a	22.7	16.7	±	7.22	a	43.1
13	Gluten free oats flour	10.7	±	6.15	a	57.3	14.1	±	4.14	a	29.3	9.25	±	6.96	a	75.2	12.4	±	0.88	a	7.1
14	Sorghum powder	36.7	±	15.7	a	42.7	7 and BLQ			a		3.63 and BLQs			a		7.67	±	2.96	a	38.6
15	Organic wheat grass juice powder	18.9	±	0.11	b	0.6	26.1	±	0.07	a	35.9	6.80	±	1.23	c	18.1	1.29 and BLQ			d	
16	Resistant wheat starch	8.19	±	1.52	b	18.6	6.8 and BLQ			b		5.76	±	0.90	b	15.6	14.9	±	0.00	a	0.0
17	Barley grass juice powder	7.47	±	2.31	a	30.9	5.15	±	0.64	a	12.4	BLQs					BLQs				
18	Resistant wheat starch	6.41	±	0.26	a	4.1	BLQs					BLQs					7.87	±	6.30	a	80.1
19	Seasoning powder	5.73	±	0.12	a	2.1	BLQs					BLQs					6.57	±	0.37	a	5.6
20	Oat flour	BLQs					BLQs					18.6	±	3.54	a	19.0	8.23 and BLQ			a	
21	Sugar babies cookie **	BLQs					BLQs					BLQs					BLQs				
22	Yeast extract powder	891	±	83.8	a, b	9.4	1539	±	818	a	53.2	176	±	28.5	b	16.2	500	±	211	b	42.3
23	Tortilla chips	494	±	53.5	b	10.8	468	±	104	b	22.2	650	±	24.9	a, b	3.8	1000	±	233	a	23.3
24	Hemp protein powder	466	±	133	a	28.5	433	±	92.0	a	21.3	104	±	25.3	b	24.2	97.3	±	31.7	b	32.5
25	Cookie	277	±	3.31	a	1.2	226	±	20.4	a	9.0	149	±	31.3	b	21.0	83.7	±	32.1	c	38.4
26	Cocoa powder	229	±	71.6	a	31.2	207	±	5.66	a, b	23.6	142	±	6.30	b	4.4	134	±	36	b	27.1
27	Bar flavor	152	±	2.63	a	1.7	134	±	18.8	a	14.0	64.6	±	9.60	b	14.9	45.9	±	0.60	b	1.3
28	Organic Alfalfa grass juice powder	121	±	6.49	a	5.3	67.7	±	2.47	b	3.7	80.2	±	6.02	b	7.5	BLQs				
29	White rice flour	92.0	±	4.39	a	4.8	79.5	±	27.5	a, b	34.6	106	±	24.0	a	22.7	35.8	±	9.22	b	25.7
30	Organic Kamut grass juice powder	78.9	±	3.22	a	4.1	67.1	±	19.3	a	28.7	59.8	±	1.92	a	3.2	6.94 and BLQ			b	
31	Barley grass juice powder	65.5	±	9.34	a	14.3	43.2	±	5.23	b	12.1	62.6	±	9.17	a	14.6	6.11	±	2.08	c	34.1
32	Organic creamy buckwheat powder	31.6	±	18.7	a	59.2	26.5	±	21.1	a	79.5	20.9	±	9.12	a	43.5	2.08	±	1.34	a	64.4

Values within each row followed by different letters are significantly different (α = 0.05). BLQ = below limit of quantification. ** Sugar babies cookie originally tested in R-Biopharm AG RIDASCREEN^®^ Gliadin assay at 10 ppm. Failure to detect gluten residues by any method in this comparative analysis is likely attributable to subsampling differences or perhaps a laboratory error in the initial analysis.

**Table 3 foods-11-00706-t003:** Analysis of gluten/wheat protein in cereal grains by the four commercial sandwich ELISA kits.

Grain Sample	Protein Content Measured by the Dumas Method (n = 3)			R-Biopharm AG RIDASCREEN® Gliadin Kit (n = 3)					Neogen Veratox® for Gliadin R5 Kit (n = 3)					Romer Labs AgraQuant® Gluten G12 Assay (n = 3)					Morinaga Wheat Protein ELISA Kit (Gliadin) (n = 3)				
	Protein ± SD, ppm			Gluten ± SD, ppm				%CV	Gluten ± SD, ppm				%CV	Gluten ± SD, ppm				%CV	Wheat Protein ± SD, ppm				%CV
**Rye, Triticale, and Barley**																							
Organic whole grain rye flour (*S. cereale*)	96,300	±	1600	3271	±	1448	a	44.3	523	±	71.3	d	13.6	811	±	94.0	c	11.6	1216	±	48.9	b	4.0
Whole Triticale grains (*× Triticosecale*)	142,900	±	3800	2619	±	1620	a	61.9	526	±	133	b	25.4	834	±	222	b	26.6	1446	±	145	a, b	10.0
Organic whole grain barley flour (*H. vugalre*)	36,300	±	1500	2188	±	775	a	35.4	577	±	253	b	43.9	2023	±	356	a	17.6	801	±	129	b	16.1
**Hexaploid wheat**																							
Whole spelt grains (*T. aestivum L. subsp. spelta (L.) Thell.*)	100,700	±	900	1785	±	858	a	48.1	629	±	193	a	30.6	737	±	21.7	a	2.9	781	±	71.0	a	9.1
Stone ground whole (hard red spring) common wheat flour (*T. aestivum L. subsp. Aestivum*)	146,300	±	1400	1224	±	547	a, b	44.7	447	±	161	c	36.1	1180	±	171	b	14.5	1513	±	67.1	a	4.4
Whole club wheat grains (*T. aestivum L. subsp. compactum (Host) Mackey*)	79,700	±	1800	388.7	±	44.9	a	11.5	147	±	15.3	c	10.5	243	±	95.1	b, c	39.2	356	±	57.5	a, b	16.2
**Tetraploid wheat**																							
Whole Emmer wheat grains (*T. turgidum L. subsp. dicoccum (Schrank ex Schübl.) Thell*)	126,000	±	1600	464.3	±	27.4	b	5.9	157	±	34.0	c	21.7	219	±	59.6	c	27.3	676	±	90.0	a	13.3
Semolina wheat flour (*T. turgidum L. subsp. durum (Desf.) Husn.*)	130,800	±	200	417.4	±	73.0	b	17.5	236	±	21.2	c	9.0	207	±	79.3	c	38.2	1031	±	196	a	19.0
Organic whole Durum wheat flour (*T. turgidum L. subsp. durum (Desf.) Husn.*)	180,600	±	86,100	414.6	±	94.2	b	22.7	172	±	20.3	c	11.8	241	±	72.6	c	30.1	1179	±	75.7	a	6.4
Whole Kamut^TM^ (Khorasan) grains (*T. turgidum L. subsp. turanicum (Jakubz.) Á. & D. Löve*)	160,800	±	1000	271.0	±	59.4	b	21.9	95.1	±	44.6	c	46.9	256	±	20.5	b	8.0	1019	±	97.2	a	9.5
**Diploid wheat**																							
Einkorn wheat kernels (*T. monococcum L. subsp. Monococcum*)	143,100	±	2800	131.1	±	16.9	b	12.9	163	±	33.0	b	20.2	97.8	±	7.86	c	8.0	485	±	83.4	a	17.2
**Oat and Sorghum**																							
Whole oat grains (*A. sativa* var. Ogle)	145,700	±	5000	BLQs					BLQs					15.4	±	1.72		11.2	BLQs				
Whole oat grains (*A. sativa* var. Jerry)	163,500	±	5000	BLQs					BLQs					15.4	±	1.55	a	10.1	5.02	±	7.38	a	147.0
Kiln roasted rolled oats (*A. sativa*)	147,900	±	1500	BLQs					BLQs					14.3	±	2.69	a	18.8	2.79	±	0.78	a	28.0
Kiln roasted whole grain oat flour (*A. sativa*)	116,600	±	3000	BLQs					BLQs					12.6	±	4.29		34.2	0.32 and BLQs				
Whole sorghum grains (*S. bicolor* var. Macia)	98,400	±	2300	BLQs					BLQs					7.02	±	3.03		43.1	BLQs				
**Spiked 1000 ppm crude wheat gluten**																							
In ground cane sugar	798,000	±	3400	1477	±	570	a	38.6	480	±	31.4	c	6.5	666	±	112	b	16.8	1151	±	160	a	13.9
In whole oat grain powder (*A. sativa* var. Ogle)	Not tested			601	±	109	a, b	18.2	518	±	188	a, b	36.3	467	±	32.9	b	7.0	757	±	115	a	15.2
In whole oat grain powder (*A. sativa* var. Jerry)	Not tested			521	±	52.4	a, b	10.1	526	±	69.5	a, b	13.2	455	±	38.7	b	8.5	577	±	45.1	a	7.8
In kiln roasted rolled oat powder (*A. sativa*)	Not tested			421	±	91.1	b	21.6	654	±	137	a	21.0	318	±	21.1	b	6.6	628	±	28.8	a	4.6
In kiln roasted whole grain oat flour	Not tested			876	±	94.8	a	10.8	607	±	64.1	b	10.5	567	±	18.2	b	3.2	937	±	136	a	14.5
In whole sorghum grain powder (*S. bicolor* var. Macia)	Not tested			67.5	±	12.6	c	18.6	653	±	47.7	a	7.3	247	±	9.40	b	3.8	8.55	±	0.47	d	5.5

Values within each row followed by different letters are significantly different (α = 0.05). BLQ = below limit of quantification.

## Data Availability

Data are available from the Food Allergy Research and Resource Progam; contact Prof. Goodman.

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
