# Peer review of "Challenges in Gluten Analysis: A Comparison of Four Commercial Sandwich ELISA Kits"

_foods, 2022, doi:10.3390/foods11050706_

Round 1

Reviewer 1 Report

The manuscript entitled "Challenges in Gluten Analysis: A Comparison of Four Commercial Sandwich ELISA Kits" is evaluated the analytical performances in commercial Gluten ELISA kits. Although similar Gluten ELISA analyses are exist, this manuscript evaluated  the gluten in various types of foods using four types of gluten ELISA.  These types of analyses are valuable for each user of ELISA kits. Thus, this manuscript should publish on research article.

Author Response

Thank you for the kind remarks about our manuscript.

Reviewer 2 Report

In this manuscript, the authors compared the quantification results of gluten content using four commercialized sandwich ELISA kits. Overall, they explained the difference in results and proposed their opinion on the application of the kits. The following concerns need to be addressed before the publication.

L30: Give the full name of Ig.

L37: Is the prevalence of 1% from globally or within the US?

L92: What is the type I method?

L179: Why did the authors choose 1,000 ppm as the spiking concentration?

L241: For Table 2, I have several questions. First, how to explain the large SD within the same sample using the same kit? For example, the tested result for raw meat using Neogen’s kit was 1694 ± 451. Second, how to compare between gluten and wheat protein, i.e., the first three kits gave gluten concentration while the last one give wheat protein concentration, what is the relationship between these two concentrations? Or if the comparison is necessary? Third, the authors gave some explanations on the divergence of the data, if so, which results should we trust?

Table 1: Why mAb could recognize more than one paratope?

L288: Will hydrolyzed gluten cause allergy?

Author Response

L30: Give the full name of Ig.

            Done

L37: Is the prevalence of 1% from globally or within the US?

            For U.S. and now clarified.

L92: What is the type I method?

            In AOAC, a Type I method is a defining method against which future methods are to be compared.  We have indicated “(defining method)” in parenthesis.

L179: Why did the authors choose 1,000 ppm as the spiking concentration?

            This selection was somewhat arbitrary.  However, these grain flours are quite high in gluten concentration and 1000 ppm was selected to avoid the need for large dilutions before analysis in the gluten ELISA kits that would have created greater, unavoidable dilution errors.  We also recognize that 1000 ppm falls into the upper range of likely, real-world cross contact scenarios.  We have insert a sentence here to justify our selection on that basis.

L241: For Table 2, I have several questions. First, how to explain the large SD within the same sample using the same kit? For example, the tested result for raw meat using Neogen’s kit was 1694 ± 451. Second, how to compare between gluten and wheat protein, i.e., the first three kits gave gluten concentration while the last one give wheat protein concentration, what is the relationship between these two concentrations? Or if the comparison is necessary? Third, the authors gave some explanations on the divergence of the data, if so, which results should we trust?

            We attribute the large SD for some samples to sub-sampling differences and  have added a parenthetical statement to clarify.  With the meat sample in particular, thorough homogenization is more difficult due to the physical nature of the sample.  If the gluten contamination happened to be some sort of small particle (we do not know that for a fact), sizeable subsampling differences would be anticipated.  This sort of variability due to sub-sampling differences has been very thoroughly examined in the case of gluten in gluten-free oats by other investigators and we hved cited other publications especially Fritz and Chen.  Part of the problem of sub-sampling differences can be attributed to the recommended use of rather small samples (0.25 g for gluten ELISA).  Most allergen ELISAs use considerably larger sample sizes (5 g).  This issue is discussed extensively in Fritz and Chen; we do not believe that additional dialogue on this point was necessary in our manuscript.

            The Morinaga kit does ostensibly calibrate their wheat ELISA in terms of “wheat protein” and not in units of “gluten” or “gliadin”.  However, in their own publication on this method (Saito et al, Ref 19), the Morinaga authors describe it as a wheat gluten ELISA.  Since our study was intended to compare commercial kits, we used their written methods and expressed the results as they recommend.  We believe that the Morinaga kit may actually measure wheat gliadin or gluten just like the other commercial kits but it is not our decision about how to express the results.  We also would have no conversion factor to use to recalculate the results from wheat protein to gluten; Morinaga would have to supply that conversion factor and they do not.

            In terms of which result to trust, this is now a major issue for gluten analysis in samples where sub-sampling differences are likely.  Law suits have been filed associated with gluten residues in gluten-free oats because occasional sub-samples test above 20 ppm.  However, we conducted multiple tests on each sample, so describing the mean and standard deviation for each sample is the honest approach to expression of the data.  Also, the variability in results is a major factor in our recommendation in the conclusions that manufacturers of gluten-free foods should aim target gluten levels well below 20 ppm if possible.

Table 1: Why mAb could recognize more than one paratope?

                Multiple paratopes can be recognized by monoclonal antibodies.  Table 1 merely provides the information on these commercial ELISA and their antibodies as supplied in publications (cited) by their manufacturers.  The antibodies will bind to similar structures that are shared to a great extent by the peptides shown in Table 1 and that was verified by the referenced studies.  We were not testing avidity in your study.  The avidities of binding to those peptide sequences could be slightly different, but sufficiently high for binding to be observed in the ELISA kit.  The specificity of binding to the repeated similar sequences in prolamin proteins is possible for the mAb.  We relied on references 25, 26, and 27 for the R5 ELISAs and reference 18 for the G12 ELISA.  The paratopes are not identified for the Morinaga polyclonal.

L288: Will hydrolyzed gluten cause allergy?

            The short answer is Yes, maybe some gluten peptides remain immunogenic and we have now inserted that word into the sentence on L288.  It is challenging to prove (or disprove) that specific peptides might provoke disease.  There is a substantial body of research and publications on this issue that would be far too complex to describe in our manuscript.

Reviewer 3 Report

For some group of consumers gluten is dangerous food ingredient. To assure the individuals safety, the legal regulations of labelling "gluten-free" food, as containing below 20 ppm of gluten has been established. Confirmation compliance of labelling with the regulatory requires applying appropriate controlling procedures as well as analytical methods. Currently for the gluten detection in complicated food matrixes mainly are used ELISA tests. The commercial ELISAs can detect gluten presence at the level of 20 ppm, but each test can be differed in format, applied antibody, specificity, sensitivity, or sample preparation method.  Also obtained results may significantly vary. The choice of appropriate reliable ELISA for gluten detection in particular sample is not easy, however crucial to ensure consumers safety. So far only R5 Mendez ELISA method is indicated by AOAC as official method for gluten analysis, but even this has some limitation. Therefore, the studies on effectiveness of ELISA tests for the determination of gluten in various foods are on interest of numerous of researchers. The submitted manuscript fits this topic very well. 

Specific comments:

The manuscript is interesting provide an advance in current knowledge, shedding new light on the selection of ELISA tests for the detection of gluten in different foods, very important issue for consumer safety. The aim was well, clearly defined.

The study was correctly designed. The material and methods were chosen and described properly in an unequivocal way. The numbers of samples as well as variety of analysed matrices were very well selected.

Obtained results together with statistical analysis are presented in two logically completed tables. Very strong, well prepared part of the manuscript is interesting discussion of results, the interpretation, discussion with other authors, clear explanation leading to conclusions

Overall merit: The authors proof that all analysed ELISA kits (R-5 Mendez method as well as other) yield comparable results and could be used interchangeable in many food matrices this expand the numbers of reliable methods of gluten free food authentication. Nevertheless, authors mentioned that the suitability of each kit should be verify before specific uses.

Author Response

Thank you for the kind remarks regarding our manuscript.

Reviewer 4 Report

The manuscript addresses a pertinent topic regarding the available methods based on ELISA to quantify gluten. The topic is not new, but the idea of determining the best method for gluten analysis is pertinent. Still, the manuscript presents some concerns.

Major comment:

One of the major concerns regards the lack of preparation of a model food containing known amounts of gluten (example model cookies), in order to allow the normalization of the sample. At the same time, it would allow drawing conclusions regarding the interference of food processing and matrix. Criteria such as accuracy and precision of each method would certainly be possible with such a model. Additionally, it could also help defining the best detection method for gluten. It would of major interest, if authors could doing a model food containing gluten covering the levels of the dynamic ranges of each kit.

Other comments.

Lines 28-33 - add the citation.

Line 82 - correct to "sensitized consumers".

Line 124 - Please provide city, country for commercial brand Sigma.

Table 1 - LOD and LOQ are not in good agreement. Please check units.

Lines 216 - sentence too confusing and repetitive. Please rephrase it.

Lines 232-234 - Do the authors know the true value of gluten within the tested samples? In order to actually determine the kit with best accuracy, it would be to test on model samples with known amounts of gluten.

Lines 234-240 - the values for precision are hard to evaluate without presenting the coefficient of variance. Please add to Table 2.

Table 3 - did the authors used the same amount of total protein for each analysis?

Line 338 - citation is not provide in the reference list. Please confirm if citation is correct.

Discussion – can the authors correlate their results with the detection capacity of each test regarding different serving size portions? Please discuss.

Author Response

Major comment:

One of the major concerns regards the lack of preparation of a model food containing known amounts of gluten (example model cookies), in order to allow the normalization of the sample. At the same time, it would allow drawing conclusions regarding the interference of food processing and matrix. Criteria such as accuracy and precision of each method would certainly be possible with such a model. Additionally, it could also help defining the best detection method for gluten. It would of major interest, if authors could doing a model food containing gluten covering the levels of the dynamic ranges of each kit.

Clearly we did not perform our comparison of these commercial ELISA kits in the manner preferred by this reviewer.  To some degree, a study using a model food matrix has previously been performed and published; we referenced that publication (Sharma et al. Reference #36).  Instead, we focused on comparison of results obtained with a variety of real-world samples of different matrix compositions.  If we had instead conducted research on one or several model foods, we would not have learned about the impact of matrix composition on results, although we agree that we would have determined the comparative accuracy and precision of the different kits in the selected matrices.  We wish the editors to consider publication of our manuscript without the need to perform additional research as suggested by this comment.

Other comments.

Lines 28-33 - add citation Costa et al 2022 (https://doi.org/10.1007/s12016-020-08810-9) or similar.

We agree that this sentence and a sentence on lines 35-37 should be referenced. Costa et al. (2022) is not an appropriate reference because it does not have a sufficiently thorough discuss on the disorders associated with gluten-containing grains.  Instead we selected Cabanillas (2020) which is now Reference 1.

Line 82 - correct to "sensitized consumers".

We disagree with this suggestion. It is a picky point but “sensitized” primarily describes the immunological process involved in IgE-mediated food allergies such as IgE-mediated wheat allergy. A consumer is sensitized when they develop IgE antibodies that specifically recognize and bind to wheat protein (or some other allergen in other food allergies). In celiac disease, the process differs. We assert that “sensitive consumers” is the proper term to cover celiac disease, nonceliac gluten sensitivity, and IgE-mediated wheat allergy.

Line 124 - Please provide city, country for commercial brand Sigma.

Done

Table 1 - LOD and LOQ are not in good agreement. Please check units.

Done.  Thank you for pointing out the error.  The Morinaga one should be ppm.

Lines 216 - sentence too confusing and repetitive. Please rephrase it.

We have attempted to rephrase the sentence to enhance clarity.

Lines 232-234 - Do the authors know the true value of gluten within the tested samples? In order to actually determine the kit with best accuracy, it would be to test on model samples with known amounts of gluten.

We do not know the true values for these real-world samples. This comment relates to the major comment that was addressed earlier.

Lines 234-240 - the values for precision are hard to evaluate without presenting the coefficient of variance. Please add to Table 2.

Done. The %CV values were added to the Table 2 although calculating %CV when analyzing only 2 sub-samples is minimal at best.

Table 3 - did the authors used the same amount of total protein for each analysis?

No, the ELISA analyses were conducted using total sample weight as instructed by the kits.

Line 338 - citation is not provide in the reference list. Please confirm if citation is correct.

Thank you for identifying this error. The Rubiero reference is now added to the reference list.

Discussion – can the authors correlate their results with the detection capacity of each test regarding different serving size portions? (Holzhauser et al 2020 https://doi.org/10.1016/j.fct.2020.111709). Please discuss.

Round 2

Reviewer 4 Report

The authors have addressed most comments.